*Perspective*

Aging, Senescence and Plasticity

# Molecular evolution of animal aging

Daniel H Nussey[1], Fabrizio d'Adda di Fagagna[2,3], Allison J Bardin [4], Helen M Blau[5], Anne Brunet[6], Dmitry V Bulavin[7], Longhua Guo[8], Eiji Hara [9], Jan Philipp Junker [10,11,12], Vera Gorbunova [13,14], Maria Mittelbrunn [15], Michael Rera [16], Jane Reznick[17], Andrei Seluanov [13,14], Björn Schumacher [17,18,19], Emma C Teeling[20], Dario Riccardo Valenzano [21], Jing Ye[22,23], Maximina H Yun [24], George A Garinis [25]✉ & Eric Gilson [7,23,26]✉

## Abstract

**Comparative biology plays a crucial role in uncovering fundamental biological mechanisms and providing evolutionary models for their variation. This approach is particularly valuable for studying aging, given the remarkable diversity in aging trajectories across the tree of life. Many evolutionary theories of aging were proposed well before the discovery of the molecular mechanisms involved, and they remain largely theoretical. Moreover, the growing number of model organisms and the expanding array of experimental and theoretical approaches used to study aging have often remained compartmentalized. As a result, integrating these diverse insights into a unified framework has become increasingly important. As a step toward this goal, this field perspective outlines general biological mechanisms that help explain the variability in aging patterns and longevity across the animal kingdom.**

**Keywords** Aging; Evolution; Development; Senescence; Environment

## Introduction

In the coming decades, our societies will face two seismic shifts: one demographic, with an increasing proportion of the human population reaching old age; the other ecological, with effects of global changes on aging dynamics of wildlife populations. Both events point to the importance of the mechanisms of aging in terms of public health and ecological sustainability. However, although aging is a universal process, its biological mechanisms, evolution, and relationships with the environment are fairly unexplored and consequently poorly understood. Therefore, there is a need to integrate the various dimensions of aging into comprehensive frameworks.

Biologically, aging can be defined as the progressive and inexorable failure of homeostasis (Partridge and Barton, 1993). Aging is accompanied at all levels of biology with an accumulation of damage associated with dysfunctional molecular, cellular and systemic processes leading to organismal aging (generating physiological and behavioral changes) and ultimately to population aging (impacting reproduction and survival, which are key for fitness and evolution). While it is well known that developmental conditions are important for adult functioning and fitness, emerging evidence indicates that they also impact aging (Blackburn and Epel, 2012). Conversely, aging can be studied as a developmental program. Thus, an emerging concept is that the root of aging lies deep in development (Gladyshev, 2021; Lu et al, 2023). Research on aging will therefore benefit from the study of developmental processes.

Despite still being an emerging field of research, biogerontology has made remarkable progress in identifying molecular principles of cellular aging over the last two decades (Lopez-Otin et al, 2023). The categorization of these principles into "hallmarks of aging" has proven useful, as experimental modulation of these hallmarks in various model organisms can alter aging trajectories. Yet, the interest of "hallmarks" for aging research is debated (Gems and de Magalhaes, 2021), and there is no single biological process or "hallmark" sufficient to explain by itself variation in aging across the tree of life. In nature, we encounter remarkable variation in lifespan and demographic aging across individuals, species,

[1]Institute of Ecology and Evolution, The University of Edinburgh School of Biological Sciences, Edinburgh, Edinburgh, UK. [2]IFOM ETS-The AIRC Institute of Molecular Oncology, Milan, Italy. [3]Institute of Molecular Genetics IGM-CNR "Luigi Luca Cavalli-Sforza", Pavia, Italy. [4]Institut Curie, PSL Research University, CNRS UMR 3215, INSERM U934, Stem Cells and Tissue Homeostasis Group, Paris 75005, France. [5]Baxter Laboratory for Stem Cell Biology, Department of Microbiology and Immunology, Stanford School of Medicine, Stanford, CA 94305-5175, USA. [6]Glenn Laboratories for the Biology of Aging, Stanford University, Stanford, CA, USA. [7]Institute for Research on Cancer and Aging of Nice (IRCAN); Université Côte d'Azur, INSERM; CNRS, Nice, France. [8]Department of Molecular Integrative Physiology, Department of Cell Developmental Biology, Institute of Gerontology, Geriatrics Center, University of Michigan, Ann Arbor, MI, USA. [9]Department of Molecular Microbiology, Research Institute for Microbial Diseases, Osaka University, Suita, Japan. [10]Max Delbrück Center for Molecular Medicine in the Helmholtz Association, Berlin Institute for Medical Systems Biology, Berlin, Germany. [11]Charité – Universitätsmedizin Berlin, Berlin, Germany. [12]German Center for Cardiovascular Research (DZHK), Site Berlin, Berlin, Germany. [13]Department of Biology, University of Rochester, Rochester, NY, USA. [14]Department of Medicine, University of Rochester Medical Center, Rochester, NY, USA. [15]Centro de Biología Molecular, Consejo Superior de Investigaciones Científicas, CSIC-UAM, Madrid, Spain. [16]Université Paris Cité, Biologie Fonctionnelle et Adaptative, Paris, France. [17]Cologne Excellence Cluster on Cellular Stress Responses in Aging-Associated Diseases (CECAD), University Hospital Cologne, University of Cologne, Cologne, Germany. [18]Institute for Genome Stability in Aging and Disease, Medical Faculty, University and University Hospital of Cologne, Cologne, Germany. [19]Department of Biology, University of Crete, Heraklion, Crete, Greece. [20]School of Biology and Environmental Science, University College Dublin, Dublin, Ireland. [21]Leibniz Institute on Aging, Fritz Lipmann Institute (FLI), Jena, Germany. [22]Department of Geriatrics, Medical Center on Aging of Shanghai Ruijin Hospital, Shanghai Jiaotong University School of Medicine, Shanghai, China. [23]Pôle Sino-Français de Recherches en Sciences du Vivant Et Génomique, RuiJin Hospital, Shanghai Jiao Tong University School of Medicine, CNRS, Inserm, Université Côte d'Azur, Shanghai, China. [24]Chinese Institutes for Medical Research, Beijing, China. [25]Institute of Molecular Biology and Biotechnology (IMBB), Foundation for Research and Technology-Hellas, Heraklion, Crete, Greece. [26]Department of Medical Genetics, Institut Hospitalo-Universitaire (IHU) RESPIRera, CHU Nice, Nice, France. ✉E-mail: garinis@imbb.forth.gr; eric.gilson@univ-cotedazur.fr
https://doi.org/10.1038/s44318-026-00725-z | Published online: 27 February 2026

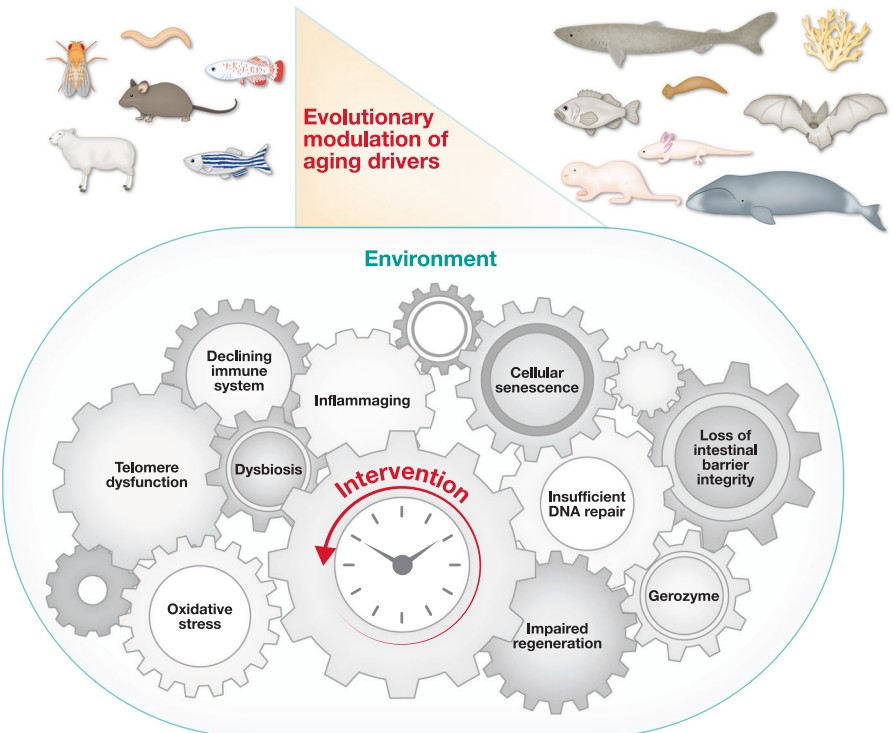

**Figure 1. Aging is fundamentally a time-dependent process (central clock) interacting with the environment (blue oval).**

Biologically, this interaction drives multiple interconnected processes (aging drivers represented by cogwheels) leading to a loss of homeostasis and increased susceptibility to a wide range of diseases, and ultimately to death. The temporal hierarchy and relative importance of these aging drivers are still poorly understood. Therefore, the arrangement of aging drivers written within the cogwheels is arbitrary. This Perspective describes how these drivers have been attenuated (anti-aging) or enhanced (pro-aging) during the evolution of a wide range of animals, thus explaining the great heterogeneity observed among them in terms of aging trajectories and longevity. The processes shown in the figure are in no way an exhaustive list of known aging drivers. Schematically, the short-lived organisms reviewed here are shown on the left, as particularly sensitive to molecular drivers, whereas the long-lived species on the right appear to have evolved strategies that delay the pro-aging effects of these drivers. Whether the mechanisms that accelerate or slow aging in animals act mainly on shared drivers, or instead reflect distinct strategies across taxa to target different drivers, remains unclear. In addition, the extent to which these evolutionary changes confer adaptive value in specific environments and ecosystems still calls for in-depth investigation through large comparative field studies, both within and among species. In any case, the anti-aging mechanisms invented during evolution are an invaluable source of bioinspiration for human interventions aimed at healthy aging (indicated by the red arrow rotating counterclockwise).

populations and space (Holand et al, 2016; Jones et al, 2014; Lemaitre et al, 2013; Nussey et al, 2013). Why has this variability evolved? Do the same "hallmarks of aging" identified as important in laboratory animals explain this variation? How do the molecular processes shaping aging vary across species and environmental conditions? What role do developmental processes and conditions play in shaping the onset and rate of aging across species? These fundamental questions remain largely unanswered. Yet, they are critical not only for advancing the biology of aging but also for designing interventions to mitigate age-related decline.

Evolutionary theory provides a powerful lens through which to understand variation in aging. Central to these theories is the idea that natural selection weakens with age (Hamilton, 1966). The presence of this so-called "selection shadow" in later life

enables senescence to evolve via two mutually non-exclusive genetic mechanisms: the accumulation of late-acting deleterious mutations via mutation/selection balance (mutation accumulation model) and the selection for alleles with positive effects on early fitness but detrimental effects in later life (antagonistic pleiotropy model) (Williams, 1957). Alongside these population-genetic frameworks, the disposable soma theory proposes that organisms evolve to allocate limited resources preferentially to reproduction rather than somatic maintenance, leading to progressive decline of repair systems and physiological aging of the body ("soma") (Kirkwood, 1977). However, these evolutionary models offer limited insight into the molecular and physiological processes shaping aging (Lemaitre et al, 2024).

Understanding why particular pathways or hallmarks matter in specific taxa, and how developmental processes interact with

environmental constraints to shape aging, requires synthesizing and comparing mechanisms identified in classical model organisms with those discovered in non-model species spanning broad phylogenetic and ecological contexts. This integrative biogerontological perspective was central to the EMBO Workshop "*Developmental Circuits in Aging*" (Nice, France, April 9–12, 2024), which focused on animal aging. This multi-author, multi-model dialog addressed key questions in biogerontology, including whether aging bears a taxonomic signature (Fig. 1). The diverse mechanisms modulating aging that were presented for a wide array of animal species indicate that aging drivers differ in both impact and temporal hierarchy among taxa. Clarifying why particular pathways or hallmarks are pivotal in some species but not in others will illuminate how genetics and environment jointly sculpt aging and support the view

that aging mechanisms constitute an evolutionary toolbox used to tune aging trajectories to environmental and ecosystem constraints. At the same time, studies of wild populations show that harsh, rapidly changing environments can accelerate aging in species such as sheep, whereas long-lived species like bats and reef-building corals often display greater resilience to environmental adversity. Together, these insights highlight long-lived organisms as powerful sources of bioinspiration for medical innovation in healthy aging, offering blueprints to enhance DNA repair, regeneration, immune function, and resilience to environmental change. The following chapters summarize complementary field views on five key aging-related processes: DNA repair, immunity, regeneration, response to unfavorable environment and models of longevity regulation.

## How to better repair DNA and protect telomeres ?

Efficient DNA repair is required for maintaining life (Schumacher et al, 2021). In parallel, maintaining functional telomeres despite their somatic erosion emerges as a key mechanism countering aging (Bernardes de Jesus et al, 2012; Rossiello et al, 2022; El Mai et al, 2023). There are multiple examples of mutations in DNA repair and telomere genes that lead to progeroid phenotypes (Armanios and Blackburn, 2012; Kyng and Bohr, 2005). However, are more efficient DNA repair and telomere maintenance associated with longer lifespan? This question is important to determine whether DNA repair and telomere pathways are relevant targets for improving human healthspan and lifespan.

In the following paragraphs, we will see how comparative analyses of long-lived vertebrates showed that enhanced DNA double-strand break repair is a recurrent evolutionary strategy to extend lifespan and healthspan. Next, work in *C. elegans* and mammalian cells identified the conserved DREAM complex as a master repressor of DNA repair genes, whose inhibition can boost genome maintenance and attenuate progeroid traits. On the telomere side, studies in mouse and zebrafish demonstrated that persistent DNA damage signaling at telomeres drives senescence and aging, and that antisense oligonucleotides targeting this signaling can ameliorate age-associated defects. Finally, studies on

replicative senescence revealed that telomere shortening promotes pericentromeric dismantling and chromatin instability during human aging.

### Improving DNA double-strand break repair, lessons from long-lived species (Vera Gorbunova and Andrei Seluanov)

Improving DNA repair is a promising strategy to counteract the deleterious effects of aging. The best solution to improve DNA repair has been provided by comparative biology studies that examined DNA repair efficiency across species with different maximum lifespans. Remarkably, analysis of DNA double-strand break (DSB) repair in 20 rodent species with maximum lifespans ranging from 3 to over 40 years revealed very strong positive correlations between the efficiency of the nonhomologous end joining pathway of repair of DSB and maximum species lifespan (Tian et al, 2019). Homologous recombination, another DSB repair mechanism, revealed an even stronger positive association with maximum species lifespan. In contrast, nucleotide excision repair showed no correlation to species longevity but a strong association with diurnal lifestyle and sun exposure. When comparing only nocturnal or burrowing rodents, there was a trend towards a positive association between nucleotide excision repair and lifespan, but it was relatively small. A large transcriptomic study comparing 24 rodent species across multiple tissues identified DSB repair as being one of the top biological pathways positively associated with species longevity (Lu et al, 2022). This result indicates that evolving longer lifespans requires upregulating the expression of factors involved in DSB repair. Studies of exceptionally long-lived species similarly revealed evolutionary pressure to improve DNA repair. This includes a study across multiple species of rockfish that showed long-lived fish species to have signs of positive selection in DNA repair genes (Kolora et al, 2021). Genomic analysis of multiple bat species that are exceptionally long-lived for their size showed positive selection of genes involved in DSB repair and homologous recombination (Zhang et al, 2013). Two recent studies analyzing longevity champions, the Greenland shark, which is the longest-lived vertebrate and the bowhead whale, the longest-lived mammal, showed positive selection in DNA repair in the

Greenland shark (Sahm et al, 2024), and exceptionally efficient DSB repair in the bowhead whale (Firsanov et al, 2025). Taken together, there is strong evidence that DSB repair shows a particularly strong association with longevity with long-lived species showing positive selection and higher expression of DSB repair genes and more efficient DNA repair process as measured in cell culture assays. These studies point to DNA double-strand break repair as being a pathway that can be targeted to improve lifespan and healthspan.

### Targeting DREAM, lessons from the nematode (Björn Schumacher)

How to target DNA damage repair to improve lifespan and healthspan? While defective DNA repair can greatly accelerate human aging, a major question in the field has been whether the reverse could be accomplished, i.e., improved DNA repair. The most striking difference between DNA repair capacities is the one distinguishing the germline and the soma. Indeed, somatic mutation rates are by at least an order of magnitude higher than germline mutation rates (Milholland et al, 2017). The distinction between somatic and germline expression of DNA repair genes is particularly striking in the *C. elegans* model, where some of the repair mechanisms are largely confined to the germline (Vermezovic et al, 2012). Many DNA repair genes are repressed by the DREAM complex, which in the nematodes was found to operate in somatic cells. DREAM mutants indeed show derepression of DNA repair genes operating in distinct repair machineries (Bujarrabal-Dueso et al, 2023). The DREAM complex is highly conserved from worms to humans. Similarly to *C. elegans* DREAM mutants, pharmacological inhibition of the DYRK1A kinase, which regulates the assembly of DREAM, leads to induction of a wide range of DNA repair genes and boosts resistance to distinct DNA damage types in human cells. Moreover, a progeroid disease model displaying retinal degeneration was shown to be protected from DNA damage built up and photoreceptor loss by DYRK1A inhibition using the natural compound harmine (Bujarrabal-Dueso et al, 2023). These results establish for the first time that DREAM operates as a master regulator of DNA repair and could serve as a pharmacological target for improving

genome maintenance, which could lower cancer risk by reducing somatic mutation accumulation and promote longevity by improving somatic genome stability (Bujar-rabal-Dueso et al, 2025).

### Protecting telomeres in mouse and zebrafish (Fabrizio d'Adda di Fagagna)

DNA damage in telomeric repeats is more challenging to repair than damage in other regions of the genome. DSB generation elicits a prompt signaling cascade that, in proliferating cells, arrests cell proliferation until DNA damage has been repaired in full. Previously, persistent DNA damage was shown to promote cellular senescence (Fumagalli et al, 2012). Although it remains unclear how some genomic DSB can persist and thus promote cellular senescence, at least one mechanism has been identified to explain their irreparability. Such mechanisms rely not on the peculiarity of the lesion per se, but rather on its genomic location. Indeed, it has been shown that DSB that happen to occur within telomeric repeats resist endogenous DNA repair activities and thus fuel persistent DNA damage signaling leading to cellular senescence (Fumagalli et al, 2012; Hewitt et al, 2012).

While it is difficult to imagine strategies to improve DNA repair within telomeric tracts, while at the same time preserving their ability to prevent chromosomal fusions (which are DNA repair events among telomeres), opportunities arise from the recently acquired ability to control the consequences of telomeric DNA damage: that is, its signaling. Indeed, DSB triggers the local synthesis of RNA which are necessary for DNA damage signaling (Fran-cia et al, 2012), and that antisense oligonu-cleotides (ASO) against them can dampen it (Michelini et al, 2017), including at telo-meres (Rossiello et al, 2017). This results in reduced cellular senescence and, in some progeric mouse models, in lifespan length-ening (Aguado et al, 2019), demonstrating the causative contributing role of telomeric DNA damage signaling to lifespan. Telo-mere biology is conserved across species, and thus it is expected that such a mechan-ism can be implemented broadly. Indeed, recently, DNA damage signaling inhibition at telomeres by ASO was shown to reduce several of the developmental, reproductive and fitness defects in *Danio rerio* (zebrafish) carrying a mutation in the telomerase (*tert*) gene (Allavena et al, 2025).

Overall, these studies suggest that DNA damage signaling, rather than DNA damage per se, controls senescence and aging. The discovery of a tool, telomeric ASO, to decouple damage from signaling will allow to explore this in vary many species— given that the biology and telomeric DNA sequence is conserved throughout vertebrates.

### Mitigating heterochromatin dismantling in human aging (Jing Ye)

Telomeres have evolved from the need to stabilize natural chromosome ends, result-ing in a particular terminal chromatin structure based on the specific recruitment of a protective protein complex named shelterin (de Lange, 2018). Interestingly, some of these telomeric factors are able to localize outside telomeric regions (Martinez et al, 2010; Simonet et al, 2011; Yang et al, 2011), where they can regulate genome stability (Mendez-Bermudez et al, 2018) and transcription (Ye et al, 2014), suggest-ing that the anti-aging functions of shelterin subunits can rely both on their telomeric and extra-telomeric functions. This is the case for TRF2, which executes global func-tions outside telomeres by mediating the late replication and facilitating fork progres-sion of pericentromeric heterochromatin (Bauwens et al, 2021; Mendez-Bermudez et al, 2020; Mendez-Bermudez et al, 2018). During replicative senescence, loss of peri-centromeric heterochromatin, DNA damage as well as a reduction in the number of pericentromeric satellite DNA repeats are caused by p53-dependent downregulation of TRF2, establishing a direct mechanistic link between telomere shortening and loss of constitutive heterochromatin during senescence (Mendez-Bermudez et al, 2022). Unpublished results also reveal that this pericentromeric "dismantling" can occur during human aging in peripheral blood mononuclear cells in parallel with TRF2 downregulation (Cheng et al, in preparation). These findings suggest the existence of a telomere-TRF2-pericentromere axis whose alteration con-tributes to aging.

### How to manage immunity?

Throughout evolution, animals have co-adapted with microbes, developing complex immune systems that both defend against pathogens and xenobiotics and maintain beneficial relationships with commensal and symbiotic microorganisms (Hanson, 2024). For example, it has been hypothesized that the use of fire links microbiome composi-tion and the evolution of human aging (Danchin, 2018): this change in microbe composition to support host health by aiding digestion would have influenced the aging process.

The immune system not only protects against infections and harmful substances but also contributes to tissue repair and to the clearance of cancer and senescent cells, making immune decline a central driver of aging. Senescent cells can escape immune surveillance by expressing PD-L1 (Pro-grammed death-ligand 1), conferring resis-tance to T cells (Wang et al, 2022a), or the ganglioside GD3 (sialic acid–containing glycosphingolipid), conferring resistance to natural killer (NK) cells (Iltis et al, 2025).

Immune complexity differs markedly between invertebrates and vertebrates: invertebrates rely on innate immunity alone, whereas vertebrates possess adaptive immunity with diverse antigen receptors and antigen-specific memory. With age, vertebrate adaptive immunity not only becomes less effective at controlling infec-tions and tumors, but also fails to clear senescent cells (Carrasco et al, 2022), and may even become autoreactive, contributing to inflammaging (Desdin-Mico et al, 2020; Mogilenko et al, 2021; Yousefzadeh et al, 2021).

Understanding how immune aging reshapes host–microbe interactions, tumor surveillance, chronic inflammation, and the clearance of damaged or senescent cells is crucial for developing strategies that enhance resilience and healthspan in later life. In the following paragraphs, we will see how age-related dysfunction of immune cells, particularly T cells, can act as a core driver of inflammaging, linking mitochon-drial defects, DAMP release, barrier disrup-tion, and senescent-cell accumulation to systemic chronic inflammation and age-related disease. Next studies revealed that persistent DNA damage generates cytosolic DNA and RNA–DNA hybrids that activate innate antiviral pathways, enabling the development of exosome-based delivery of nucleases to clear these nucleic acids, dampen inflammatory signaling, and pro-tect prematurely aging mice, thus pointing to a therapeutic route to curb DNA damage-driven inflammation. Finally, work on the microbiota-senescence–immunity axis

demonstrated that chronic bacterial exposure induces senescence in ileal B cells, reduces IgA production, and drives gut dysbiosis, establishing a feedback loop in which senescence, immune imbalance, and microbiota changes jointly erode homeostasis with age.

## Targeting inflammaging through the immune system (Maria Mittelbrunn)

Inflammaging, defined as a state of chronic low-grade inflammation in older individuals, has emerged as a key contributor to age-related diseases, including cardiovascular conditions, metabolic disorders, and cognitive decline—together representing major global causes of mortality. This process is characterized by elevated levels of pro-inflammatory cytokines, chemokines, and other mediators, which serve not only as hallmarks of inflammaging but also as indicators of biological age. Although multiple triggers of inflammaging have been identified—including external factors such as pathogens and pollution, as well as internal factors like mitochondrial dysfunction, microbiota dysbiosis, and the release of damage-associated molecular patterns (DAMPs)—recent findings suggest that the intrinsic deterioration of immune cells, particularly T cells, may represent a central and integrative cause of inflammaging (Aranda et al, 2024). T cells are among the cells that change the most with aging (Escrig-Larena and Mittelbrunn, 2025). Age-associated mitochondrial dysfunction in T cells promotes a pro-inflammatory and self-aggressive phenotype, along with a loss of their protective functions. This dysfunction may contribute to the release of DAMPs, disruption of intestinal barrier integrity, and accumulation of senescent cells—altogether fueling systemic inflammaging (Gabande-Rodriguez et al, 2019).

By integrating approaches based on immunometabolism, cellular therapies, and pharmacological and nutritional interventions, the aim is to restore immune cell function—not only to delay immune aging, but also to mitigate inflammaging and reduce the burden of age-associated diseases.

## Targeting cytosolic DNA to counteract inflammation in the mouse (George A. Garinis)

Aging is increasingly recognized as a process driven not only by genetic and metabolic decline but also by chronic inflammation rooted in cellular damage. As we age, persistent DNA damage accumulates, triggering inflammation—the body's initial immune response to foreign pathogens and irritants. Over time, DNA damage-induced inflammation intensifies, driven by immune system feedback and amplification loops. This escalation can lead to cellular dysfunction, tissue degeneration, and metabolic complications. Using a unique series of mice with an inborn defect in DNA repair, which causes premature aging, DNA damage was shown to progressively accumulate with age, activating immune DNA-sensing mechanisms (Karakasilioti et al, 2013; Niotis et al, 2026). This establishes a direct link between innate immune signaling and the DNA damage response. In $Ercc1^{-/-}$ mice, DNA damage-driven R-loops (three-stranded nucleic acid structures formed during transcription when a nascent RNA hybridizes back to its DNA template) contribute to the release and build-up of single-stranded DNA in the cytoplasm, triggering a viral-like immune response in both progeroid and naturally aged pancreata (Chatzidoukaki et al, 2021). Further investigation revealed that microglial cells in these mice accumulate double-stranded DNA (dsDNA) and chromatin fragments in the cytosol, which are sensed by the immune system, prompting a similar viral-like response in the brain (Arvanitaki et al, 2024). These cytosolic DNA fragments are packaged into extracellular vesicles (EVs) released by microglia, which deliver their dsDNA cargo to interferon-responsive neurons, ultimately leading to cell death. In more recent work, it was demonstrated that cells with a transcription elongation defect due to the loss of the transcription elongation factor TFIIS (encoded by the gene *TCEA1* gene) exhibit stalled RNA polymerase II (RNAPII) at oxidative DNA damage sites, impaired transcription, R-loop accumulation, telomere uncapping, chromatin bridges, and genome instability, ultimately resulting in cellular senescence (Siametis et al, 2024). R-loops at telomeres were shown to cause the release of telomeric DNA fragments into the cytoplasm, triggering a viral-like immune response. TFIIS-defective cells release EVs carrying these telomeric DNA fragments, which target neighboring cells and induce cellular senescence.

To mitigate this inflammation and remove cytoplasmic nucleic acid fragments in damaged cells, we developed an exosome-based therapeutic strategy to deliver recombinant nucleases to inflamed cell were developed, reducing viral-like inflammatory signals in vivo (Arvanitaki et al, 2024; Chatzidoukaki et al, 2021; Siametis et al, 2024). Moving toward physiological applications, empty or nuclease-loaded EVs were administered to DNA repair-deficient mice via intraperitoneal injection or intranasal delivery for five consecutive days. This treatment substantially removed cytoplasmic DNAs and RNA–DNA hybrids from the pancreas, liver, and brain microglial cells of $Ercc1^{-/-}$ mice. It also significantly reduced pro-inflammatory cytokine levels, IFN1 levels, and Purkinje cell death in the cerebellum of these rapidly aging mice (Arvanitaki et al, 2024). Together, our findings reveal a causal mechanism leading to tissue inflammation and offer a promising therapeutic strategy against age-related degenerative diseases.

## Targeting the cross-talk microbiota-senescence-immunity (Eiji Hara)

The microbiome's impact on aging has been suggested to be mediated at least in part through the immune system, influencing the body's ability to fight infections and inflammatory conditions, which are pivotal in the aging process (Popkes and Valenzano, 2020). However, what triggers the changes in the gut microbiota with aging remains a key question. Age-related dysbiosis may result from dietary changes, lifestyle alterations, and decreased gastrointestinal function. However, recent evidence suggests that changes in the production of Immunoglobulin A (IgA), which targets gut bacteria, might contribute to the dysregulation of gut microbiota composition during aging (Macpherson et al, 2018). Recent findings revealed that chronic, age-associated exposure to commensal gut bacteria under standard SPF conditions, induces cellular senescence in ileal germinal center (GC) B cells, reducing IgA production, thereby changing the composition of gut microbiota in aged mice, thereby causing gut microbiota imbalance (Kawamoto et al, 2023). Therefore, the crosstalk between gut microbiota and cellular senescence appears to trigger immune system abnormalities and dysbiosis, contributing to the loss of homeostatic mechanisms associated with aging.

## How to regenerate better?

Organs consist of cells with a large diversity of specialized roles. A fundamental question

is how these cells mount a coordinated response in space and time to maintain or restore organ function after perturbation. Organismal aging is generally associated with impaired maintenance and restoration capacities. Although high capacity for regeneration is common among invertebrates, some being capable of whole-body regeneration, in the majority of vertebrates, tissues such as the liver and skin can regenerate to maintain or restore original size and function after damage, whereas others, such as the heart, have a very limited regenerative capacity (Cai et al, 2023; Poss, 2010; Tanaka and Reddien, 2011). Given the vital role of tissue regeneration in organismal health and disease, deciphering the mechanisms underlying tissue regeneration following damage and identifying barriers to improved tissue regeneration in older age remain important goals.

The studies in the following paragraphs will show how different organisms link regeneration to aging in distinct ways. In planarians, whole-body regeneration can reverse molecular and cellular hallmarks of aging upon injuries. In mouse skeletal muscle, age-related upregulation of the prostaglandin-degrading enzyme 15-PGDH reduces PGE2 and weakens regeneration, while pharmacological inhibition of 15-PGDH restores youthful repair and strength, identifying a druggable "gerozyme" axis. Studies in *Drosophila* will highlight that the rapid accumulation of somatic mutations in adult stem cells undermines tissue renewal and likely accelerates aging. Next, zebrafish heart studies revealed that efficient regeneration relies on both cardiomyocyte plasticity and transient pro-regenerative fibroblasts and immune responses that are largely absent in adult mammals. Finally, in axolotls, lifelong regeneration coincides with negligible senescence, efficient clearance of senescent cells, sustained telomere maintenance, and atypical epigenetic aging, positioning this species as a key model to understand how durable regenerative capacity can buffer or delay aging.

## Whole-body regeneration and rejuvenation in planarians (Longhua Guo)

Freshwater planarians are champions of whole-body regeneration. They are widely considered "immortal" for their life history, regenerative capacity, and lack of natural death for decades in laboratory culture.

Recent work established the sexual lineage of *Schmidtea mediterranea* as a model system for aging and regeneration research (Dai et al, 2025). These sexual planarians produce progeny in a week after mating, which allows accurate recording of birthdays, and hence, ages. Surprisingly, these "immortal" beings do not have an extremely slow aging rate. Between 1 and 2 years, these animals reach reproductive senescence, lose vigor and show increased oxidative stress. Single-cell mRNA sequencing revealed that aging in planarians is associated with loss of neuronal and muscle mass, as well as changes in classical aging-related molecular pathways (e.g., insulin/mTORC1 signaling, oxidative phosphorylation, immune response, and mRNA splicing). Remarkably, these age-associated changes can all be reversed in animals after injury and regeneration (Dai et al, 2025). In other words, regeneration in sexual planarians led to the reversal of tissue aging, a potential mechanism for their extremely long lifespan.

The regenerative power of planarians is driven by resident adult stem cells (Reddien et al, 2005). In planarians, around 3 years old, no loss of adult stem cells was observed (Dai et al, 2025). While age-associated molecular changes were identified in differentiated tissues (e.g., intestine, epidermal), changes in stem cells were minimal. These observations suggest adult stem cells in planarians age at a much slower rate than differentiated tissues, providing a potential mechanism for their extreme longevity and rejuvenation capacity. This work also suggests rejuvenation in mammals may be achieved by strategies to remove aged tissues and replace them with new and young tissues.

## Skeletal muscle regeneration in the mouse by targeting a gerozyme (Helen M Blau)

Skeletal muscle is a highly plastic tissue capable of extensive regeneration owing to tissue-resident stem cells known as muscle stem cells (MuSCs). The efficiency of the muscle repair process declines with aging, causing tissue loss and dysfunction. Prior studies showed that muscle stem cells (MuSCs) meet the quintessential definition of a stem cell—self-renewal and differentiation—and are essential to skeletal muscle regeneration (Buckingham, 2007; Sacco et al, 2008). 15-PGDH, the prostaglandin-degrading enzyme, was recently found to be a pivotal molecular determinant of aging

and named a "gerozyme" (Bakooshli et al, 2023). This discovery builds on prior work showing that PGE2, a component of the body's natural healing mechanism, is crucial for muscle regeneration. Results show that if PGE2 signaling to MuSCs is disrupted, muscle repair after injury is impaired and strength declines (Ho et al, 2017). In good agreement, studies in other tissues, such as colon, liver, and hematopoiesis, show that PGE2 boosts stem cell function regeneration (Ho et al, 2022; Koh et al, 2025; Smith et al, 2021; Zhang et al, 2015). PGE2 levels decline with aging due to a mere 2-fold increase in the expression of the gerozyme 15-PGDH. However, this decline can be countered by boosting PGE2 levels by reducing the activity of the gerozyme via treatment with a small molecule inhibitor (PGDHi), which acts via multiple mechanisms, including augmenting MuSC function (Wang et al, 2022b; Wang et al, 2025). In addition to its effects on stem cells, PGE2 boosts mitochondrial number and function in mature muscle fibers and causes a remarkable remodeling of the aged muscle tissue architecture (Palla et al, 2021). Strikingly, modulating PGE2 to levels seen in young mice via PGDHi treatment restores neuromuscular connections at synapses of aged mouse muscles in a manner previously only seen with exercise (Bakooshli et al, 2023). This is due to PGE2's effects in promoting motor neuron sprouting and survival.

15-PGDH appears to be a pivotal molecular determinant of muscle aging. Overexpression of the gerozyme 15-PGDH in young mouse muscles leads to a reduction in tissue PGE2 levels and causes muscles to shrink and weaken. Conversely, inhibition of 15-PGDH (15-PGDHi) causes aged mouse muscles to increase strength by 10–15% within 1 month, suggesting that this represents a promising therapeutic approach (Palla et al, 2021). Thus, restoring PGE2 levels via 15-PGDHi is a "triple threat": it rejuvenates muscle stem cells and myofibers and restores neuromuscular junctions (Bakooshli et al, 2023; Palla et al, 2021; Wang et al, 2025). Notably, 15-PGDHi has been shown by others to blunt or reverse features of aging in the brain using a mouse model of Alzheimer's and in the blood by restoring the lymphoid bias typical of young (Koh et al, 2025; Smith et al, 2021). These unexpected findings suggest that targeting this gerozyme can ameliorate muscle strength and other aged

tissue functions and mitigate the ravages of age on our brain and brawn.

## Stem cell aging in *Drosophila* (Allison J Bardin)

Stem cell DNA damage has been proposed to be one cause for the functional decline of tissue renewal during aging (Sharpless and DePinho, 2007). *Drosophila* represents a powerful model in which to address questions of stem cell aging since it has stem cell-renewed adult tissues such as the germlines, associated soma, and the digestive tract, that undergo physiological decline over a relatively short time scale (5–7 weeks). Studies have demonstrated that spontaneous somatic mutations arise surprisingly frequently in adult intestinal stem cells and increase dramatically during the aging process (Riddiford et al, 2021; Siudeja et al, 2015). The nature of the spontaneous mutations is diverse and ranges from large structural variants, and deletions to insertion of mobile transposable elements as well as loss of heterozygosity (Al Zouabi et al, 2023; Rubanova et al, 2025; Siudeja et al, 2021). Consistent with rapid aging correlating with increased rate of somatic mutation, a study examining somatic mutation rates across diverse mammals demonstrated an inverse relationship with lifespans (Cagan et al, 2022). Moreover, recent work has begun to unravel how somatic mutations in stem cells impact other aging hallmarks, such as inflammation during aging, to drive clonal expansion as illustrated in the blood (Jakobsen et al, 2024). Altogether, these findings support the notion that the rate of somatic mutations in stem cells plays a major role in aging.

As stem cells have unique properties, including specialized niches, responses to the environment (Al Zouabi and Bardin, 2020), future studies aimed at understanding and ameliorating negative physiological consequences of stem cell DNA damage and mutations are needed.

## Heart regeneration in zebrafish (Jean Philipp Junker)

Heart injury in adult mammals typically leads to permanent scarring. However, the adult zebrafish heart regenerates efficiently after injury, making zebrafish a powerful model for studying mechanisms of heart regeneration (Gonzalez-Rosa et al, 2017;

Poss et al, 2002). Seminal work has shown that the heart muscle regenerates via dedifferentiation and proliferation of cardiomyocytes (Jopling et al, 2010; Kikuchi et al, 2010), but the contribution of non-myocytes to heart regeneration remains less well understood. Cryoinjury to the ventricle of the heart was shown to trigger activation of a transient fibroblast population characterized by expression of *col12a1a*, which is completely absent in the healthy heart (Hu et al, 2022). After targeted ablation of this transient fibroblast population, heart regeneration was significantly reduced, demonstrating the pro-regenerative function of *col12a1a* fibroblasts. Massively parallel lineage tracing based on CRISPR/Cas9-induced genetic scars revealed the epicardial origin of this transient fibroblast population (Hu et al, 2022). In unpublished work (Mintcheva et al, *in preparation*), RNA metabolic labeling by single-cell SLAM-seq (Holler et al, 2021) was used to directly measure the immediate response to heart injury within the first few hours, with the goal to identify the sentinel cells that induce the following cascade of regenerative events. SLAM-seq allows measurement of two time points per gene in each single cell: old unlabeled RNA that was transcribed before injury, and new labeled RNA molecules transcribed after injury. This approach leads to a considerable improvement in the signal-to-noise ratio, revealing that activation of three damage-response pathways (Toll-like, C-type lectin and NOD-like receptor signaling) in macrophages and endocardium is the first response of the zebrafish heart to injury. In summary, spatio-temporal single-cell omics has the potential to reveal the mechanisms and interactions that underlie the cell state changes that drive regeneration.

## Axolotl insights into regeneration and aging (Maximina H Yun)

Salamanders are the evolutionarily closest organisms to humans capable of regenerating extensive sections of their body plan, including parts of their eyes, heart, brain, spinal cord, tail and limbs as adults (Cox et al, 2019). In these organisms, regeneration has two salient features. First, instead of relying exclusively on stem cells, the progenitors for the new structure are often obtained through dedifferentiation and transdifferentiation of mature adult cells (Lin et al, 2021; Yu et al, 2023). Second, in

contrast to other vertebrates, their regenerative capacity by and large does not decline with time (Yun, 2021). Furthermore, salamanders exhibit extraordinary longevity (Sousounis et al, 2014), an apparent lack of age-related decays throughout lifespan (Yun, 2021), cancer resistance and defiance of the Gompertz-Makeham law of mortality (Cayuela et al, 2019; Reinke et al, 2022). As such, they have long been considered species of negligible senescence (Yun, 2021). However, it is unclear to what extent do salamanders age, and whether their exceptional regenerative capacity -and in particular their ability to exert reversals of cell fate- underlie their apparent lack of age-related decay.

By establishing aging biomarkers and systematically exploring the manifestation of aging hallmarks through lifespan and regeneration in the experimentally-tractable axolotl (*Ambystoma mexicanum*), it was shown that these animals are able to circumvent a major hallmark of aging, namely the accumulation of senescent cells with age (Yun et al, 2015), a phenomenon that may be associated with a highly efficient immune surveillance mechanism acting during regeneration (Yu et al, 2023). Further, salamander studies uncovered the upregulation of mechanisms of telomere maintenance in regenerative contexts (Yu et al, 2022), suggesting that these may be active throughout the lifespan. Indeed, unpublished results indicate that axolotls do not show telomere attrition up to geriatric ages (animals over 13 years old, 10–13 years old is the average lifespan), a phenomenon accompanied by active telomerase activity in several tissues across the lifespan. Results also suggest that axolotls exhibit distinct aging patterns at the organ-level, exemplified by a lack of thymus involution until the edge of their lifespan (c. 10 years of age, even though axolotls sexually mature at 10–12 months) in contrast to other vertebrates where thymus involution begins early in life, contributing to immunosenescence (Liang et al, 2022). At the molecular level, they provide new tools to explore epigenetic changes with age, in particular DNA methylome alterations, which indicate that axolotls deviate from the typical patterns of age-related epigenetic modifications seen in senescent species (Haluza et al, 2024) and provide means to study the impact of regenerative processes on biological age. Collectively, these considerations put forward the axolotl as an emerging model for uncovering the basis of

negligible senescence and its links to regeneration, of relevance to current theories of aging and the design of therapeutic strategies towards ameliorating aging.

## How to deal with an unfavorable environment?

From the environmental point of view, studies on the contribution of aging biology to the dynamics of animal populations and biodiversity are still in their infancy. Understanding the aging trajectories of wildlife populations and their modifications in response to global changes has become an essential scientific question (Gaillard and Lemaitre, 2019; Nussey, 2025). The future answers will certainly have to be considered in sustainable development strategies.

The studies in the following paragraphs will illustrate how diverse environments shape aging strategies. Long-term studies of wild Soay sheep showed how early-life and lifetime ecological challenges influence actuarial and reproductive senescence, with a minority of "super sheep" maintaining high performance into old age. Reef-building corals revealed that telomere and oxidative-stress regulation can be either highly environment-sensitive (short-lived coral) or buffered (long-lived coral), linking molecular aging traits to habitat variability. Naked mole-rats demonstrate how paedomorphic, fetal-like cardiac structure and metabolism can be tuned to hypoxic subterranean burrows while conferring exceptional longevity and disease resistance. The African turquoise killifish adopts the opposite strategy, evolving a compressed life cycle and diapause to survive ephemeral ponds, and providing a fast, genomically tractable model in which both environment and microbiota (via lifespan-extending young-to-old microbiome transfers) modulate vertebrate aging. Finally, bats combine high metabolic rates with extreme longevity and low cancer incidence, maintaining telomeres, mitochondrial integrity, microbiome diversity, and balanced immunity in changing environments, making them outstanding models of environmentally robust healthy aging.

### Super sheep (Daniel H Nussey)

Many long-term field studies of wild vertebrates now represent powerful—albeit underused and underappreciated—model systems for our understanding of how complex fluctuating natural environments shape the aging process. Crucially, because these studies track known individuals continuously from birth to death, they allow us to test how challenging environments, experienced at different points across the lifetime, shape the accumulation of molecular hallmarks of aging and impact an individual's ability to maintain homeostasis in later life.

The long-term study of Soay sheep on St Kilda is a good example of such studies, and recent work on this system has sought to bring together detailed ecological understanding with new work to measure markers of key hallmarks of aging. The Soay sheep are a very primitive breed of domestic sheep that have lived unmanaged for several millennia on the remotest archipelago in the United Kingdom. Since 1985, the sheep population resident in the Village Bay area of the largest island (Hirta) have been individually caught and marked at birth and then closely monitored throughout their lifetimes (Clutton-Brock, 2004). This study, now approaching its 40th year, has collected complete records of the life histories of >8000 individuals across 14 generations.

Just like in humans, there is dramatic variation in the pattern of aging among individuals in natural populations. In the Soay sheep system, there are strong signals of actuarial and reproductive senescence from around 6 years of aging onwards, and also evidence that body weight and home range area decline in later life in both sexes (Froy et al, 2018; Hayward et al, 2015). Yet, there are some remarkable examples of "super sheep" (the wild equivalent of human centenarians, perhaps), that have achieved exceptional longevity, and sustained high reproductive performance and social dominance into ripe old age.

Long-term field study systems like the Soay sheep will allow us to address outstanding questions, including: (1) Are there trade-offs between resilience to different kinds of environment pressure (e.g. maintaining immune function against parasites, food limitation, predation) in later life? (2) How important are developmental versus current or accumulating environmental challenges in shaping variation in aging rates? (3) How does developmental rate, which is known to be very sensitive to the environment, shape later patterns of aging?

## Coral model of extreme longevity in changing environments (Eric Gilson)

Telomeres are environment-sensitive regulators of health and aging. To address the question of whether mechanisms regulating telomeres evolve differently in organisms acquiring specific life history traits, reef-building corals was used as model organisms. Indeed, these animals are particularly suited here due to their ectothermic metabolism, which makes them more plastic to environmental factors, their sessile lifestyle in the adult stage, which prevents escape from environmental stress, allowing for tracking the effects of past events, and their broad range of life-history traits, including short and long-lived species sharing the same habitat.

Thanks to the *Tara Pacific* Expedition (Planes et al, 2019), over a large geographical range across the Pacific Ocean, telomere DNA length variation was compared between the massive *Porites* spp. (spp. for *species plurimae*) that are slow growing, stress-resistant, and long-lived, with estimated life expectancies of >600 years, and the branched *Pocillopora* spp. that are fast growing, sensitive to bleaching, and short-lived with estimated average colony ages of a few decades. Telomere DNA length variation was largely explained by historical patterns of sea surface temperatures (SST) in both coral genera, albeit with marked differences. *Pocillopora* spp. telomeres were sensitive to seasonal temperature variation, whereas *Porites* spp. Telomere DNA length was slightly positively correlated with the presence/prevalence of past heat waves (Rouan et al, 2023). These results reveal complex relationships between telomere and environment that go beyond the current view that telomere DNA length decreases upon environmental stress. Moreover, utilizing the same samples, *Pocillopora* spp. have oxidative stress response biomarkers (protein carbonylation and total antioxidant capacities) structured by the geography and the environment, while *Porites* spp. have a more stable oxidative stress response across the Pacific Ocean (Porro et al, 2023). Overall, these results reveal that two aging regulators, telomere and oxidative stress response, are environment-sensitive in the short-lived *Pocillopora* spp. and more environment-insensitive in the long-lived *Porites* spp. They suggest that specific mechanisms regulating aging in response to

environmental variations contribute to the different longevity and stress-resistance properties encountered among reef-building corals. Additionally, they allow us to think about how to target telomere maintenance and oxidative stress response for preventing and treating the adverse effects of heat waves on health and aging in humans.

## Paedomorphy to live in an unfavorable environment, lessons from the naked mole-rat (Jane Reznick)

Paedomorphy, maintenance of juvenile traits throughout life, is a feature of certain species often inhabiting unfavorable environments, including certain salamanders, cavefish, naked mole-rat, and even human (Gould, 1977). Notably, paedomorphy can arise through slowed somatic development (neoteny), a process frequently linked to increased disease resilience and extended lifespan.

The naked mole-rat (*Heterocephalus glaber*) is a small rodent species with extraordinary longevity living over 35 years, about five-fold greater than predicted allometrically for a 40-gram rodent (Ruby et al, 2018). Age-associated chronic diseases, including cancer, neurodegeneration, and cardiovascular disease, are extremely rare (Oka et al, 2023), with similar incidences in both young and old individuals. Being eusocial and living in large groups of up to 300 individuals in subterranean burrows imposes harsh environmental challenges, including hypoxia, hypercapnia, and sparsely available food and water. Naked mole-rats display a panel of diverse juvenile features across morphological, molecular, and hormonal parameters (Buffenstein et al, 2020).

Signatures of neoteny in the naked mole-rat heart have been observed across various parameters, including cardiac contractile machinery, which closely resemble the myofilament protein signature of fetal mice (Grimes et al, 2017) (e.g., fetal forms of MHC proteins and troponin isoforms) as well as a higher percentage of mononucleated cardiomyocytes (Gan et al, 2019). Moreover, we showed recently that naked mole-rats utilize fetal cardiac metabolism, replacing the typically adult fatty acid utilization for a unique type of carbohydrate metabolism largely dependent on glycogen (Bundgaard et al, 2024). Taken together, the distinct juvenile-like state of the naked mole-rat myocardium may reflect functional and metabolic optimization for the

hypoxic, hypercapnic, and acidic milieu of their deep underground burrows, which simultaneously enable heightened repair of injured cardiomyocytes.

Retention of paedomorphic states appears to be an adaptive response to ecological pressures, promoting a coordinated program of enhanced tissue repair, regenerative capacity, metabolic flexibility, and greater resistance to stress. These traits are essential for survival in challenging environments and simultaneously offer protection against age-related disease.

## How to quickly develop, age and die to adapt to an extreme environment ? Lessons from killifish (Anne Brunet)

A main challenge in experimentally assessing the mechanisms of aging in a complex vertebrate organism is the issue of scale. Aging studies in classical experimental models (mice or zebrafish) are limited due to the long lifespan of these species (~3 and ~5 years, respectively) and their low-throughput nature. To address this challenge, the African turquoise killifish *Nothobranchius furzeri* has been adopted as a new model for aging research (Boos et al, 2024; Harel and Brunet, 2015; Valdesalici and Cellerino, 2003).

The killifish originates from Zimbabwe and Mozambique and lives in ephemeral ponds that last only ~4 months each year. This fish species has evolved two remarkable adaptations to this extreme habitat. First, it has a naturally compressed life cycle – it grows fast, reproduces fast, and ages fast, which is extremely helpful to study aging in the laboratory. Second, it has a form of "suspended animation" called diapause (Hu and Brunet, 2018), in which developed embryos with several tissues synchronously pause development, resulting in protection against damage that normally accumulates with the passage of time. In the laboratory, the killifish can be used as a powerful model organism to study both vertebrate aging and suspended animation. The killifish has many attractive characteristics for a model system: it is the shortest-lived vertebrate that can reproduce in captivity; it recapitulates typical age-dependent phenotypes and pathologies, notably decline in cognitive function and neurodegeneration with age; it is a vertebrate, and it has organs and tissues that are unique to vertebrates and critical during human aging, including an adaptive

immune system, bones, and a cardiovascular system.

Several tools have been established to study the killifish. The genome of the African killifish was sequenced, assembled, and annotated (Harel et al, 2015; Valenzano et al, 2015; Reichwald et al, 2015). Importantly, a variety of genome-editing tools have been developed for this fish, including a CRISPR/Cas9 genome-editing platform to knock out killifish genes (Harel et al, 2016). More recently, a CRISPR/Cas9 knockin pipeline have also been established, thereby allowing cell- and tissue-specific expression (Bedbrook et al, 2023; Krug et al, 2023). Finally, several groups have established transgenesis methods to overexpress genes in the killifish (Allard et al, 2013; Hartmann and Englert, 2012). These technological advances have transformed this shortest-lived vertebrate into a genetically accessible model.

Promising epigenomics, transcriptomics, proteomics, and lipidomics datasets are already available in the killifish, both in the context of aging and "suspended animation" (diapause). Transcriptomics (RNA-seq) datasets of tissues during aging are available (Baumgart et al, 2016; Xu et al, 2023). Transcriptomics, chromatin accessibility, and lipidomics datasets have also been generated for the state of diapause (Hu et al, 2020; Singh et al, 2024). Finally, the field has generated proteomic datasets for both total proteins and aggregated proteins in different tissues (notably the brain) from young and old killifish (Chen et al, 2024; Harel et al, 2024), and these datasets can serve to benchmark characteristics of biological age. Collectively, the benefits of the killifish, including its short lifespan, well-defined and conserved aging characteristics, low costs, abundant progeny, genetic tractability, and large-scale datasets to benchmark aging, make it an ideal model for conducting high-throughput studies of aging and longevity. Together, these data have revealed that the killifish recapitulates several changes with age observed in longer-lived species (increased inflammation, increased protein aggregation), corroborating its compressed life in an adverse environment.

## Ecology and evolution of microbiota in killifish (Dario Valenzano)

Studies using various model organisms have underscored alterations in the gut

microbiota as a critical factor contributing to aging and age-related diseases (Ghosh et al, 2022). Interestingly, species-specific microbiomes have emerged as a crucial interface between organisms and their environment. Indeed, host-specific microbial communities modulate nutrient absorption (Basolo et al, 2020), immune system development (Rooks and Garrett, 2016), locomotion (Schretter et al, 2018), cognition, intra- and inter-species communication, and metabolic buffering.

In killifish (*Nothobranchius furzeri*), one of the shortest-lived vertebrate models, D.R. Valenzano's team showed that transplanting gut content from young to middle-aged recipients extends lifespan, and retards age-dependent locomotor decline, suggesting a causal link between microbiome composition and the aging process (Smith et al, 2017).

If microbiota appears to contribute to health and aging in individuals, whether microbiota composition can explain differences in metabolism, longevity and aging among species with different lifespans is largely unexplored. For instance, do ultra-long-lived species owe their extreme longevity and specialized biology to their microbial partners? Can microbes help their hosts access otherwise un-accessible environments, such as extreme ocean depths, mountain tops, or oxygen-poor environments? Can we harness the elevated metabolic flexibility of complex microbial communities to buffer aging, improve resilience and extend lifespan in otherwise short-lived species?

### Wild bats as new models of extended healthspan in a changing environment (Emma Teeling)

To solve the human ageing problem, researchers have suggested that "*we need to be more creative*" (de Magalhaes, 2021). A creative, alternative approach is to explore aging in species that are even more "aging-resistant" than humans and have naturally evolved longer healthspans. By far the most successful mammals in this regard are the bats (Austad, 2010; Munshi-South and Wilkinson, 2010). Despite their small size and self-powered flight being the most metabolically demanding type of locomotion (Jebb et al, 2018), bats can live on average 3.5 times longer than non-flying mammals after correcting for body size (Wilkinson and Adams, 2019). In fact, bats represent 18 of the 19 mammals that have a

longer lifespan than humans, given body size (Austad, 2010). Furthermore, they exhibit limited senescence and negligible tumorigenesis (Foley et al, 2018; Teeling et al, 2018), suggesting that bats have evolved unique mechanisms to slow down expected aging (Athar et al, 2025; Cooper et al, 2024; Gorbunova et al, 2020).

However, logistically it is difficult to study bats in an aging context, as most are only found in the wild, are protected, are too small for nonlethal sampling and not easily maintained in captivity (Foley et al, 2018). By overcoming these hurdles, the first molecular ageing studies of wild bats revealed that members of the longest-lived genera of bats (e.g., *Myotis*) exhibit unique "pro-longevity" adaptations and do not show expected levels of age-related accumulated damage nor inflammation (Cooper et al, 2024). Rather they maintain telomere length (Foley et al, 2018; Foley et al, 2020), mitochondrial integrity (Jebb et al, 2018), microbiome diversity (Hughes et al, 2018), and immune balance with age (Huang et al, 2019), which is further supported by genomic studies (Jebb et al, 2020; Morales et al, 2025). This is associated with increasing levels of DNA repair, autophagy, and tumor suppression with age (Huang et al, 2019). In contrast, shorter-lived bats (e.g., *Molossus molossus*) exhibit higher-levels of "anti-longevity" gene expression (Huang et al, 2020). DNA methylation studies comparing shorter- vs. longer-lived bats implicate innate immunity and cancer suppression as major drivers of bat longevity (Wilkinson et al, 2021). With >1487 recognized bat species (Simmons and Cirranello, 2025) comprising 20% of all living mammals, and the majority of bats showing extended longevity while surviving in ever-changing environments, bats are extraordinary model systems in which to discover the molecular basis and evolution of longer healthspans with relevance to humans.

## Models of longevity regulation

We have seen significant improvement in our healthspan over the past century, bringing us closer to the absolute lifespan of 120 years observed in centenarians (Crimmins, 2015; Vaupel, 1997; Vaupel et al, 2021). Whether humans can live beyond 120 years, however, remains largely uncertain. The current view is that to reach this age target, one should interfere with common causes of aging, in recent years defined as hallmarks of aging (Lopez-Otin

et al, 2023). Nevertheless, new models are needed for an in-depth understanding of the mechanisms limiting lifespan and healthspan.

The two following studies focusing on when and how to intervene in aging processes can serve as models to understand longevity regulation. The first describes how cellular senescence can be a double-edged sword: clearing p16$^{high}$ senescent cells can rejuvenate tissues and even enhance the benefits of partial epigenetic reprogramming, yet some senescent cells contribute to defense and tissue protection, so future strategies must selectively remove harmful senescence while preserving beneficial forms and maintaining detoxification capacity. The "Smurf" model in *Drosophila* proposes that aging comprises an initial, relatively quiet phase followed by a late phase marked by loss of intestinal barrier integrity, multiorgan failure, and transcriptome-wide aging signatures; delaying the transition into this Smurf phase extends lifespan. This two-phase framework suggests that, beyond improving external conditions, the most effective longevity interventions may be those that act early to postpone entry into the high-mortality, systemic-failure phase rather than trying to reverse damage once that phase is established.

### Distinguishing good from bad cellular senescence, lessons from the mouse (Dmitry Bulavin)

There are now multiple examples where attenuation of various hallmarks of aging can improve specific pathological conditions and, in some cases, extend healthspan (Panchin et al, 2024). Among these, targeting cellular senescence has emerged as one of the most widely studied strategies in recent years (Childs et al, 2017). However, the issue remains complex, as certain types of senescent cells perform important beneficial functions (Born et al, 2023; de Magalhaes, 2024; Grosse et al, 2020; Reyes et al, 2022)—for instance, as part of a defense mechanism known as disease tolerance (Triana-Martinez et al, 2024). Thus, selectively attenuating deleterious senescence while preserving or enhancing beneficial forms may represent an important challenge in the years to come.

Another promising strategy for improving healthspan, which has existed for decades but is receiving renewed attention,

involves preserving the body's detoxification functions. This includes both protecting organs that contribute to increased toxin burden (such as the aging-related leaky gut) and enhancing detoxification processes (notably in the liver). Ultimately, the functional decline of these tissues leads to a higher load of circulating toxins, which disproportionately affect sensitive cell types, including vascular endothelial cells—thereby laying the groundwork for two of the most common causes of death: heart attack and stroke (Grosse and Bulavin, 2020).

To push the boundaries even further, more disruptive interventions have recently been proposed, such as partial epigenetic reprogramming using the four Yamanaka factors (4FPR) (Ocampo et al, 2016). A key concern, however, is whether the rejuvenating effects of epigenetic reprogramming could be noticeably blunted in fully aged tissues due to age-related changes in the epigenome, persistent chronic inflammation, tissue fibrosis, and other degenerative conditions. In this context, our recent findings have underscored the critical role of the continuous accumulation of p16[High] senescent cells as one of the driver of age-related tissue deterioration, which in turn impairs the efficacy of 4FPR. Encouragingly, removing these senescent cells has been shown to substantially enhance 4FPR outcomes (Grigorash et al, 2023).

Overall, the complexity of the aging process and the limitations of current strategies make it clear that there is no quick fix for extending healthspan and lifespan. Nonetheless, the road ahead promises significant—and potentially transformative—discoveries.

### Model to postpone the death of old age (Michael Rera)

Using *Drosophila melanogaster* provided evidence that the aging hallmarks, at least at the transcriptional level (Frenk and Houseley, 2018), are mostly markers of a late phase of life characterized by a loss of controlled intestinal permeability, detected, among other physiological changes, by a small non-toxic blue food dye (the Smurf phase in reference to the comic-book series featuring small blue humanoid creatures) (Rera et al, 2012). This Smurf phase was described as evolutionary conserved across *Drosophila* species, nematodes, zebrafish and mice (Cansell et al, 2025; Dambroise et al, 2016). Across all these models, aging is

made up of two consecutive and necessary phases. A first one, where one cannot detect most of the hallmarks of aging, besides a time-dependent increasing gene expression variability. This is followed by a second one, associated with all the transcriptional hallmarks of aging, physiological alterations, organ failure (i.e., intestinal barrier function and Malpighian tubule—the fly's kidney—activity (Livingston et al, 2020)) and death. This novel model can help us identify the leading causes of aging. Indeed, most of the known "aging genes" are dysregulated in the Smurf phase and delaying the Smurf entry increases lifespan (Zane et al, 2023).

How could this apply to humans? In agreement with this possibility, past reports showed an increased intestinal permeability in critically ill patients (Angarita et al, 2019; Harris et al, 1992). The dramatic increase in life and health span observed in humans starting in the early 20th century is due to the improvement of medicine, socio-economic and environmental conditions. Can we expect to identify genetic or pharmaceutical interventions leading to a postponing of death beyond what can be obtained by improving environmental conditions? It is unlikely since these improvements mainly affected child mortality, barely affecting the maximum observed lifespan, if not simply by increasing the pool size of individuals likely to reach it. But if aging following two consecutive phases, with the distinct properties discussed above, is conserved in humans, the molecular properties of these phases suggest that the first phase would be the one to target, to postpone the entry into the high-mortality/multiorgan failure/ transcriptome-wide changes Smurf phase that seems too late to reverse.

## Peer review information

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

## Acknowledgements

We thank EMBO for making the "developmental circuits in aging" workshop possible. We also thank Marie-Joseph Giraud Panis, Florentin Remot and Aaron Mendez-Bermudez for their critical reading, as well as Ge Gao for the drawing in Fig. 1. We apologize for the omission of relevant articles due to space limitations. The work in the laboratory of the authors is supported by the following grants: **EG**: ANR grants (CORALFORCE, TELOTHERM and TELOMEMORY), FRM team, "program labellisé Fondation ARC", Horizon 2020 Marie Curie ITN "HealthAge" and the Inserm programs AGEMED and INTERAGING. **FdAdF**: ERC advanced grant TELORNAGING—835103. ERC POC TELOVACCINE—101113229. AIRC-IG 30471. AIRC-IG 21762.Telethon GMR23T2007. Progetti di Ricerca di Interesse Nazionale (PRIN) 2020CXFL4T. Progetti di Ricerca di Interesse Nazionale (PRIN) 2022R7LH5T. Next Generation EU, in the context of the National Recovery and Resilience Plan, Investment PE8 Project Age-It. Investment CN3 National Center for Gene Therapy and Drugs based on RNA Technology. **DRV**: DFG CRC1310. Carl-Zeiss Foundation "microbAlome, AI-based microbiome analysis in aging". **LG**: Glenn Foundation for Medical Research, AFAR, The Pew Charitable Trusts and the National Institute on Aging (DP2AG093207, R21AG084959). **AB**: Program "Investissements d'Avenir" launched by the French Government and implemented by ANR (ANR-20-CE13-0013_01, ANR-23CE13-0024-01, and ANR 23CE15003802). **EH**: AMED grants (JP25gm1710004h0004 and JP23zf0127008h0002), JSPS grant (JP25H00443), JST grant (JPMJMS2022) and Takeda Science Foundation. **MHY**: Deutsche Forschungsgemeinschaft (22137416, 450807335, and 497658823) grants. Chinese Institutes for Medical Research, Technische Universität Dresden, Center for Regenerative Therapies Dresden, Faunsome Inc. and ALTOS Labs funds. **GAG**: Horizon-Widera-Access-Pathways to Synergies 2023 "TRIAD" (101158508); the Horizon-Widera-2023-Talents-01-01: ERA Chairs "RACE" (1011803090); Pharmathen S.A. (PAR00863) research funds, Greece 2.0, National Recovery and Resilience Plan Flagship program TAEDR-0535850. **AS** and **VG**: US National Institute on Aging grants. **AS**: Hevolution grant. **VG**: Impetus grant. **JPJ**: ERC CoG HEART_STATES – 101043364, and German Research Foundation (DFG) grant 458913362. **MM**: European Research Council (ERC-2021-CoG 101044248-Let T Be), and by Spanish Ministerio de Ciencia e Innovación (PID2022-141169OB-I00) grants. **JY**: National Natural Science Foundation of China (grant numbers 82225018), National Key Research and Development Program of China (grant numbers 2024YFA0918701). **JR**: ERC starting grant - METAMOLE 851653 **MR**: ANR ADAGIO (ANR-20-CE44-0010), CNRS, FRM and iMPT.

## Author contributions

**Daniel H Nussey**: Writing—original draft; Writing—review and editing. **Fabrizio d'Adda di Fagagna**: Writing—original draft; Writing—review and editing. **Allison J Bardin**: Writing—original draft; Writing—review and editing. **Helen M Blau**: Writing—original draft; Writing—review and editing. **Anne Brunet**: Writing—original draft; Writing—review and editing. **Dmitry V Bulavin**: Writing—original draft; Writing—review and editing. **Longhua Guo**: Writing—original draft; Writing—review and editing. **Eiji Hara**: Writing—original draft; Writing—review and editing. **Jan Philipp Junker**: Writing—original draft; Writing—review and editing. **Vera Gorbunova**: Writing—original draft; Writing—review and editing. **Maria Mittelbrunn**: Writing—original draft; Writing—review and editing. **Michael Rera**: Writing—original draft; Writing—review and editing. **Jane Reznick**: Writing—original draft; Writing—review and editing. **Andrei Seluanov**: Writing—original draft; Writing—review and editing. **Björn Schumacher**: Writing—original draft; Writing—review and editing. **Emma C Teeling**: Writing—original draft; Writing—review and editing. **Dario Riccardo Valenzano**: Writing—original draft; Writing—review and editing. **Jing Ye**: Writing—original draft; Writing—review and editing. **Maximina H Yun**: Writing—original draft; Writing—review and editing. **George A Garinis**: Writing—original draft; Writing—review and editing. **Eric Gilson**: Writing—original draft; Writing—review and editing.

## Disclosure and competing interests statement

The authors declare no competing interests.

