## [Peer Review File · The EMBO Journal]

Molecular evolution of animal aging

Daniel Nussey, Fabrizio d'Adda di Fagagna, Allison Bardin, Helen M. Blau, Anne Brunet, Dmitry BULAVIN, Longhua Guo, Eiji Hara, Jan Philipp Junker, Vera Gorbunova, María Mittelbrunn, Michael Rera, Jane Reznick, Andrei Seluanov, Björn Schumacher, Emma Teeling, Dario Riccardo Valenzano, Jing Ye, Maximina Yun, George Garinis, and Eric Gilson

Corresponding authors: Eric Gilson (Eric.Gilson@unice.fr) , George Garinis (garinis@imbb.forth.gr)

Review Timeline:

Submission Date:	21st Jul 25
Editorial Decision:	10th Oct 25
Revision Received:	24th Jan 26
Accepted:	27th Jan 26

Editor: Daniel Klimmeck

Transaction Report:

Dear Eric and colleagues,

Thank you again for submitting your review manuscript for consideration by the EMBO Journal, and also for your patience with our feedback, which got delayed due to the busy time of the year. Your manuscript has now been seen by three colleagues with expertise in evolutionary ageing research and related biology, whose comments are enclosed below.

As you will see, the experts much appreciate the piece, i.p. the breadth of angles taken and and diversity longevity models covered, as well as the different scales considered, going from molecular all the way to systemic and inter-individual. At a more critical side they feel that the structure of the text needs improvement, and at times the thematic focus and scope of the individual elements requires better alignment. We have discussed their feedback in detail here in the team, and overall find it to be constructive and relevant as they should well mirror the reception by our broad readership. In order to arrive at a revised intuitive version, we hence propose to restructure the text, and also change to the more multiview-oriented format of a perspective article, going back to annotating the individual paragraphs to the respective co-authors. Together with shortening intro and concluding remarks, and bundling them possibly into a preface, we believe this should allow arriving at a compelling version without a major rewrite.

I hope you find the comments to be useful and above approach would make sense to you, too. I suggest to have a brief call during the next days to align views on next steps.

Looking forward to hearing back from you on this.

Best wishes,

Daniel

Daniel Klimmeck, PhD
Senior Editor
The EMBO Journal

Referee #1:

There is a stunning level of diversity among species, and this holds true not only for the exquisite morphological differences we see in the natural world, and the fascinating diversity in how species make a living, but also when it comes to patterns of aging. I think the goal of this paper is to make the point that by considering the diversity of ways in which different long-lived species have solved the problem of how to live long and healthy lives, we might better understand the fundamental mechanisms of healthy aging in ourselves. As I turned to this paper, I thought how the promise of this paper stands out as an exciting complementary one to the highly cited set of papers by López-Otin and colleagues explaining the hallmarks of aging, and so could be both interesting and impactful.

The paper is the outcome of a 2024 EMBO meeting, "Developmental Circuits in Aging", and includes an outstanding group of co-authors. The premise of the article stated up front is that it will "(synthesize) insights from their studies, highlighting mechanisms that influence longevity across species and the lessons they offer for promoting healthy human aging." This is a lofty and worthwhile goal. Unfortunately, as the paper is currently constructed, it does not fulfill its promises. While the stories told here are terrific and well-written, the synthesis, which is critical, is insufficient.

The individual components of the manuscript are generally well-written, and full of interesting stories about systems and species that point to potential ways to ensure healthy long lifespan. But there is little structure, and no thread that connects one paragraph to the next, nor one section to the next, and so no argument for the reader to follow from start to finish. It felt like I was just bouncing from one story to the next, with no clear idea of where I was being led, nor why. There is also a some imbalance among the sections. Some focus immediately on molecular details (and often in ways that will leave non-molecular biologists lost), while others keep a much higher-level perspective. And some of those molecular sections refer to a single results by the author, but miss the larger overview to provide context within which to think about that result, and what studies should follow based on that result. It just wasn't clear how these pieces fit together, why these pieces were chosen and not others, etc. Moreover, there didn't seem to be a single author or perhaps two who had gone through to try to create a coherent manuscript.

The paper's summary tries to put all of the diverse work presented in the paper into a more integrated context. But I am afraid it is too late, as the reader is unlikely to have made it this far. The current version of this manuscript reads as though each person at the meeting contributed a few paragraphs about their area of interest and expertise, and then these components were pasted together, with some integrative thinking brought in at the end. As a reader, I want to know up front what to expect, and then to see the arguments and connections unfold as I read through the manuscript. I believe such a thing is possible here, and would definitely be worthwhile, but in its current form has not yet been accomplished.

I have specific comments on individual sections, but I will hold back from providing them here, as I am guessing that much will change—a wholesale revision of the manuscript is needed to bring it to the level where it could achieve its full, impactful potential.

Referee #2:

In Nussey et al, the authors briefly summarize some of the mechanisms of aging that have been discovered across species. The authors were all participants in a EMBO workshop on comparative aging, and the paper provides insights into the different talks presented in the workshop. I found the paper overall easy to read and of interest to aging biologists. However, I do have some issues/questions that are listed below in no particular order.

-The introduction sets up the paper as if the studies will be on comparative or non-model species, but several sections are on mice and *Drosophila*. I might make it a little clearer in the introduction that the paper will provide insights into evolution/comparative biology as well as some novel mechanisms of aging discovered in model organisms.

-As the paper was written in sections by different authors, there are some grammar issues in certain sections and flow issues section to section. I would make sure that one of the authors does a thorough edit at the end to make sure the sections cohesively work together.

-It is a little confusing how some sections are written about very specific studies while others are general overviews of a model. I think setting up the framework of the paper a little more in the introduction will help with this.

-Some sections use first person "our", others do not, which is confusing since this is a paper written by many different authors. Similarly, "aging" is used throughout but there are a couple instances of "ageing". Please keep it consistent.

-What is a "geriatric" age for an axolotl?

-"DDR" is not defined in the telomeres section.

-It is confusing that the axolotl is discussed in the regeneration section but not in the paedomorphy section, when they may be one of the strongest examples.

-In the "cytosolic DNA" section, define R-loops, TFIIS, RNAPII. In addition, this section was difficult to read/understand compared to other sections, and I would suggest revising to make it clearer.

-In the immune section, please define PD-L1 and GD3.

Referee #3:

EMBOJ-2025-121926: Insights into the molecular evolution of animal aging

General summary and opinion

The authors present a diverse and in-depth overview of the variability in patterns and processes of aging across the animal

kingdom and the molecular mechanisms underpinning them. They attempt to unify themes from evolutionary biology, developmental biology, ecology, and biomedical science in order to explore how aging differs across species and environments, and how the study of these differences may hold insights with relevance for human healthy aging. This paper has something for everyone, with an admirable breadth of taxonomic coverage spanning detailed mechanisms in model organisms to bigger-picture discussions of pro-longevity strategies in emerging model systems. It also covers a huge breadth of mechanistic scales, from DNA to cells to individuals to populations, and offers interesting and topical perspectives on how these may interact with the environment in a changing world. I particularly appreciated the emphasis on highlighting differences rather than similarities as an unconventional approach towards learning from these diverse model systems.

However, I found the structure of the manuscript quite confusing, and felt that overall the review lacked flow, often feeling like a collection of separate abstracts. It also felt at times as though authors were writing with different scopes for the review in mind - meaning that the paper lacked a coherent throughline, making it difficult for a reader to follow along with. This is particularly important given the wide potential readership of this paper - it may be picked up by a molecular biologist who works on the *Drosophila* endoplasmic reticulum, or by an ecologist who spends most of their time mist-netting bats. I think the authors would do well to think of both of these potential readers (and more) when considering their individual sections, to make sure it is understandable to a wide readership curious about the biology of aging. I think these issues could be remedied fairly easily, with some restructuring and fleshing out, including adding some summaries at the end of each section introduction to help signpost the reader. Below I outline my suggestions for improving the paper to enhance readability and ensure maximum impact.

Major concerns

1. My most major concern lies with the structure and thematic flow of the paper. I appreciate that the authors faced a herculean task in bringing together such diverse strands of research, and that they attempted an original synthesis rather than opting for something more pedestrian, such as organising by model system. However, I am not convinced that this is the best way to structure the paper. Some subsections also do not appear to really fit the overall section they have been placed in, making their placement feel a bit forced; for example, the "ecology and evolution of microbiota" section barely mentions the environment, except as a short hypothetical question. In some cases, this can be easily fixed; for example, I think the section on killifish ("How to quickly develop, age and die to adapt to an extreme environment"), which currently does not place much significance on their "unfavorable environment", would make far more sense if some of the detail on molecular and -omic resources in this model were trimmed to make space for further details of their diapause state as an adaptation to their harsh ephemeral pond environments. In other cases, I think the best way forward would be for the subsection to be switched to a different section entirely, although I don't think this is possible with the current available options.

Other arguments for restructuring include the fact that some sections are far longer than others, and the lack of logical flow between them (tissue regeneration DNA repair unfavorable environment infection and inflammation regulating death). For example, might it be more logical to begin with e.g. accrual of age-related damage (e.g. the DNA repair sections) / inflammation before then moving on to strategies for tissue regeneration? In particular, I am not convinced by the "regulating death" grouping; how are these two subsections linked, and how do they relate to the title "how to regulate death"? The section introduction suggests that they are linked because they discuss living beyond the current maximum human lifespan, which isn't really directly discussed in their content. It also suggests that they are linked by their discussion of pro-longevity interventions which relate to known hallmarks of aging - which is certainly true, but does not distinguish this section from any other sections of the review, which also deal with these themes. To clarify, I found each of these subsections individually very interesting, but I am not sure I agree with their grouping together in a section of their own. In general, I think a broad-scale restructuring would significantly enhance the manuscript.

2. The paper lacks cohesion. Sections do not seem to flow in a very intuitive way, and this can really hamper the reader's progression through the paper. I think this could be remedied quite easily; for example, a simple and impactful solution would be to add a few sentences at the end of each section introduction that mention what the ensuing subsections contain, and suggest how they interrelate. It would also be a nice touch to have subsections within the same section reference each other, so readers can understand how the sections are logically linked. It would also be very simple to edit some of the subsection titles such that they better reflect the section they are in; for example, changing "Stem cell aging" to "DNA Damage as a Driver of Stem Cell Aging", etc.

3. The papers reads as if authors are writing with different scopes for the manuscript in mind, and this makes the manuscript feel inconsistent. My impression based on the abstract and introduction is that the aims of the review are as follows: a) to unify themes from aging research in the context of evolution, development, and ecology, b) to explore diverse aging patterns across species / environments, and the molecular mechanisms underpinning these, and c) to highlight pro-longevity mechanisms across the animal kingdom that may be of relevance for healthy ageing in humans. Many of the subsections do indeed address all of these, but some, for example, read as a "pitch" for a particular model system with little or no reference to the underlying mechanisms driving their particular aging pattern. Some, on the other hand, are quite heavy on mechanistic detail but lack any meaningful reference back to the bigger picture. It would be useful to ensure that all authors are aware of the intended aims such that any sections that diverge too far from these may be revised for consistency.

4. Finally, I appreciate that the authors all work on vastly different systems, on very different scales, and in disparate disciplines. This diversity is of course one of the main strengths of the manuscript. However, as stated previously, the authors need to bear in mind the very broad potential readership of this paper, and ensure that their sections are written such that any biologist could

reasonably understand them. For example, given the comparative nature of the paper, some of the more mechanistic subsections should signpost the reader to their taxonomic location; while it is not necessarily of importance that the muscle stem cells discussed are in mice, it is useful for a reader to know that we have moved to mice when it is immediately preceded by a section about planarians. Similarly, some of mechanism-focused subsections would benefit from spending more time introducing some of the molecular players involved so as not to leave behind some of the biologists for whom these names are largely meaningless.

Minor concerns

5. I appreciate the short introductions at the beginning of each section of the review, and find them to be an extremely helpful way to guide the reader through the manuscript with enough background to understand the relevance of each subsection. However, some of the introductions were much shorter than others, and left a lot to be desired; for example, I did not find the "How to regulate death?" introduction very illuminating (although in any case, as I have stated previously, I found the grouping of this section problematic), and also found the "how to better repair DNA" introduction lacking, especially considering how long this section is. It would be useful, for example, to discuss how DNA damage accumulates with age and why, and to explain that most organisms have endogenous repair mechanisms for this, etc. etc. It is also missing a reference for "multiple examples of mutations in DNA repair genes that lead to progeroid phenotypes". In addition, I think that these introductions might be a useful place to outline some other relevant work in the field, as the subsections often focus specifically on the author's own work. This is not a complaint; I understand that breadth must be sacrificed for depth, and there is great merit in describing fine mechanistic details in a review such as this. However, the introductions could be a useful place to reference other relevant work for interested readers.

6. I enjoyed the introduction. It was very well-written, providing some original perspectives on the imminent shifts in global demography and ecology and therefore justifying the call for better integration of disciplines of aging research. It gave a very detailed yet accessible background to the molecular and evolutionary biology of aging. However, it would be useful to include some reference to the disposable soma theory of aging alongside the developmental theory, as the "other" leading physiological theory of aging. It feels like something of an omission to exclude it, particularly if this manuscript is picked up by biologists unfamiliar with evolutionary theories of aging.

7. I also very much enjoyed the conclusion, in particular the neat synthesis of the topics of the paper into new questions and insights. However, it did feel like these largely summarised and referred to the sections on the comparative biology of aging, with a focus on non-model systems and the insights to be gained from these, and somewhat glossed over many other more mechanistic sections of the review. It would be nice to see these incorporated into the concluding remarks, as they obviously formed a large part of review.

8. Considering that many people use reviews to direct research plans, it would be useful to see some more explicit suggestions for future research avenues within each subsection.

9. I liked Figure 1, but I am not sure how much it adds to the argument for the use of the Soay sheep as a model for aging in the wild, and I also think that its status as the only figure in the manuscript linked to a particular subsection confers it disproportionate significance. Unless the other authors have figures to add, I am not sure if it is worth retaining.

10. I think Figure 2 has a lot going for it. I really like the conceptualisation of these drivers of aging as cogs that can turn to spin the clock of aging faster. I also like the idea that these could be manipulated or pharmacologically targeted to reverse this. However, I am not sure how clear this metaphor is, considering that cogs are often used as symbols of biological mechanisms, and clocks are often used to represent aging - so this at first glance could just look like a cloud of aging mechanisms. I think this would be helped by adjusting the graphics such that the cogs actually interlock, and making the aging clock more integrated into its cog, such that the actual mechanical metaphor is clear. I also think it would be useful to make the brown "reverse aging" arrow more obvious, perhaps with a label (e.g. "pharmacological interventions", or an icon of a syringe, etc.), and perhaps with some other kind of progression arrow coming from the long-lived species, such that it is clear that your implication is that using these insights could lead to anti-aging strategies. I am also not sure if superimposing the diagram over the silhouetted grey age progression in the background is necessary, and I think removing it would reduce clutter in the image. I also think it would help to, at relevant points in the review, mention what these "drivers" of aging are such that the reader is primed for this figure at the end - I think the first time this word is used is in the conclusion.

Additional non-essential suggestions

11. P13 - explain what "thymus involution" is and why it is relevant for aging?

12. P15 - somatic mutations arise frequently in *Drosophila* intestinal stem cells, and somatic mutation rates across mammals correlate inversely with lifespan - but is there any evidence that these are associated with aging in *Drosophila*?

13. P17 - Capitalise DREAM in the title

14. P19 - DDR - DNA damage response? Define this acronym

15. P20 - IFN-responsive neurons - clarify that this refers to interferon

16. P25 - the final sentence seems like a big jump to go from corals to humans; can you suggest a way in which this research might link to studies on heat waves and telomere length in humans?

17. P26 - define neoteny (and distinguish it from pedomorphy)

18. P26 - "besides metabolism" - this phrase comes before metabolism has been mentioned for the first time
19. P35 - prolonged presence of gut bacteria - what is the context for this? Is this common, was this an artificial manipulation?

Referee

#1:

There is a stunning level of diversity among species, and this holds true not only for the exquisite morphological differences we see in the natural world, and the fascinating diversity in how species make a living, but also when it comes to patterns of aging. I think the goal of this paper is to make the point that by considering the diversity of ways in which different long-lived species have solved the problem of how to live long and healthy lives, we might better understand the fundamental mechanisms of healthy aging in ourselves. As I turned to this paper, I thought how the promise of this paper stands out as an exciting complementary one to the highly cited set of papers by López-Otin and colleagues explaining the hallmarks of aging, and so could be both interesting and impactful.

The paper is the outcome of a 2024 EMBO meeting, "Developmental Circuits in Aging", and includes an outstanding group of co-authors. The premise of the article stated up front is that it will "(synthesize) insights from their studies, highlighting mechanisms that influence longevity across species and the lessons they offer for promoting healthy human aging." This is a lofty and worthwhile goal. Unfortunately, as the paper is currently constructed, it does not fulfill its promises. While the stories told here are terrific and well-written, the synthesis, which is critical, is insufficient.

The individual components of the manuscript are generally well-written, and full of interesting stories about systems and species that point to potential ways to ensure healthy long lifespan. But there is little structure, and no thread that connects one paragraph to the next, nor one section to the next, and so no argument for the reader to follow from start to finish. It felt like I was just bouncing from one story to the next, with no clear idea of where I was being led, nor why. There is also a some imbalance among the sections. Some focus immediately on molecular details (and often in ways that will leave non-molecular biologists lost), while others keep a much higher-level perspective. And some of those molecular sections refer to a single results by the author, but miss the larger overview to provide context within which to think about that result, and what studies should follow based on that result. It just wasn't clear how these pieces fit together, why these pieces were chosen and not others, etc. Moreover, there didn't seem to be a single author or perhaps two who had gone through to try to create a coherent manuscript.

The paper's summary tries to put all of the diverse work presented in the paper into a more integrated context. But I am afraid it is too late, as the reader is unlikely to have made it this far. The current version of this manuscript reads as though each person at the meeting contributed a few paragraphs about their area of interest and expertise, and then these components were pasted together, with some integrative thinking brought in at the end. As a reader, I want to know up front what to expect, and then to see the arguments and connections unfold as I read through the manuscript. I believe such a thing is possible here, and would definitely be worthwhile, but in its current form has not yet been accomplished.

I have specific comments on individual sections, but I will hold back from providing them here, as I am guessing that much will change—a wholesale revision of the manuscript is needed to bring it to the level where it could achieve its full, impactful potential.

We thank the reviewer for the positive assessment of the interest of the review's topic and for the useful comments on the structure of the article. We agree that the "Conclusion" section, and the lessons that can be drawn from the different findings,

appeared too late in the manuscript. Therefore, in the revised version, we have combined the Introduction and Conclusion into a single "Preamble" section, which more clearly explains the logic behind the sequence of paragraphs describing each author's discoveries and highlights the lessons that can be drawn from such a multi-author, multi-model perspective. To clarify that each paragraph corresponds to the studies conducted by a specific author, we have added the name of the author responsible for writing that paragraph.

Referee #2:

In Nussey et al, the authors briefly summarize some of the mechanisms of aging that have been discovered across species. The authors were all participants in a EMBO workshop on comparative aging, and the paper provides insights into the different talks presented in the workshop. I found the paper overall easy to read and of interest to aging biologists. However, I do have some issues/questions that are listed below in no particular order.

-The introduction sets up the paper as if the studies will be on comparative or non-model species, but several sections are on mice and *Drosophila*. I might make it a little clearer in the introduction that the paper will provide insights into evolution/comparative biology as well as some novel mechanisms of aging discovered in model organisms.

Thank you for this comment. You are right that this article presents a broad comparison of aging mechanisms, combining model and non-model animals. We have clarified this point in the revised preamble.

-As the paper was written in sections by different authors, there are some grammar issues in certain sections and flow issues section to section. I would make sure that one of the authors does a thorough edit at the end to make sure the sections cohesively work together.

The editing of the article has been thoroughly revised

-It is a little confusing how some sections are written about very specific studies while others are general overviews of a model. I think setting up the framework of the paper a little more in the introduction will help with this.

Each section presenting the aging mechanisms discussed in the authors' paragraphs begins with a short introductory paragraph that situates the mechanism within the broader context of our current understanding of ageing, which has been improved in the revised version.

-Some sections use first person "our", others do not, which is confusing since this is a paper written by many different authors. Similarly, "aging" is used throughout but there are a couple instances of "ageing". Please keep it consistent.

This point has been clarified in the revised version by adding the name of the author responsible for each paragraph

-What is a "geriatric" age for an axolotl?

Geriatric age' refers to animals over 13 years old (10-13 yo is the average lifespan).

- "DDR" is not defined in the telomeres section.

corrected

- It is confusing that the axolotl is discussed in the regeneration section but not in the paedomorphy section, when they may be one of the strongest examples. You are right that these are two examples of paedomorphy. However, the paragraph on the naked mole-rat (J. Reznick) is focused on a metabolic adaptation to the subterranean environment, thus relevant to the section on the response to unfavorable environment while the paragraph on axolotl is focused on regeneration mechanism (M/ Yun).

- In the "cytosolic DNA" section, define R-loops, TFIIS, RNAPII. In addition, this section was difficult to read/understand compared to other sections, and I would suggest revising to make it clearer.

The terms have been better defined. In the revised version, we move this paragraph to the section on immunity managing, making it clearer in this context.

- In the immune section, please define PD-L1 and GD3.

Done

Referee #3:

EMBOJ-2025-121926: Insights into the molecular evolution of animal aging

General summary and opinion

The authors present a diverse and in-depth overview of the variability in patterns and processes of aging across the animal kingdom and the molecular mechanisms underpinning them. They attempt to unify themes from evolutionary biology, developmental biology, ecology, and biomedical science in order to explore how aging differs across species and environments, and how the study of these differences may hold insights with relevance for human healthy aging. This paper has something for everyone, with an admirable breadth of taxonomic coverage spanning detailed mechanisms in model organisms to bigger-picture discussions of pro-longevity strategies in emerging model systems. It also covers a huge breadth of mechanistic scales, from DNA to cells to individuals to populations, and offers interesting and topical perspectives on how these may interact with the environment in a changing world. I particularly appreciated the emphasis on highlighting differences rather than similarities as an unconventional approach towards learning from these diverse model systems.

However, I found the structure of the manuscript quite confusing, and felt that overall the review lacked flow, often feeling like a collection of separate abstracts. It also felt at times as though authors were writing with different scopes for the review in mind - meaning that the paper lacked a coherent throughline, making it difficult for a reader to follow along with. This is particularly important given the wide potential readership of this paper - it may be picked up by a molecular biologist who works on the *Drosophila* endoplasmic reticulum, or by an ecologist who spends most of their time mist-netting bats. I think the authors would do well to think of both of these potential readers (and more) when considering their individual sections, to make sure it is understandable to a wide readership curious about the biology of aging. I think these issues could be remedied fairly easily, with some restructuring and fleshing out,

including adding some summaries at the end of each section introduction to help signpost the reader. Below I outline my suggestions for improving the paper to enhance readability and ensure maximum impact.

We thank the reviewer for the positive assessment of the interest of the review's topic and for the useful comments on the structure of the article. In the revised version, we have combined the Introduction and Conclusion into a single "Preamble" section, which more clearly explains the logic behind the sequence of paragraphs describing each author's discoveries and highlights the lessons that can be drawn from such a multi-author, multi-model perspective. To clarify that each paragraph corresponds to the studies conducted by a specific author, we have added the name of the author responsible for writing that paragraph.

Major concerns

1. My most major concern lies with the structure and thematic flow of the paper. I appreciate that the authors faced a herculean task in bringing together such diverse strands of research, and that they attempted an original synthesis rather than opting for something more pedestrian, such as organising by model system. However, I am not convinced that this is the best way to structure the paper. Some subsections also do not appear to really fit the overall section they have been placed in, making their placement feel a bit forced; for example, the "ecology and evolution of microbiota" section barely mentions the environment, except as a short hypothetical question. In some cases, this can be easily fixed; for example, I think the section on killifish ("How to quickly develop, age and die to adapt to an extreme environment"), which currently does not place much significance on their "unfavorable environment", would make far more sense if some of the detail on molecular and -omic resources in this model were trimmed to make space for further details of their diapause state as an adaptation to their harsh ephemeral pond environments. In other cases, I think the best way forward would be for the subsection to be switched to a different section entirely, although I don't think this is possible with the current available options. We agree that the focus of each author's paragraph is somewhat disparate, but this reflects the intended format of a multi-author perspective. Nonetheless, it seems appropriate to keep Anne Brunet's killifish chapter in the "environment" section, as a major interest of this model is precisely to understand how an organism has adapted to an unfavorable environment.

Other arguments for restructuring include the fact that some sections are far longer than others, and the lack of logical flow between them (tissue regeneration \diamond DNA repair \diamond unfavorable environment \diamond infection and inflammation \diamond regulating death). For example, might it be more logical to begin with e.g. accrual of age-related damage (e.g. the DNA repair sections) / inflammation before then moving on to strategies for tissue regeneration?

We appreciate the comment of the reviewer and have change the order of the sections in the revised version, as suggested.

In particular, I am not convinced by the "regulating death" grouping; how are these two subsections linked, and how do they relate to the title "how to regulate death"? The section introduction suggests that they are linked because they discuss living beyond the current maximum human lifespan, which isn't really directly discussed in their content. It also suggests that they are linked by their discussion of pro-longevity interventions which relate to known hallmarks of aging - which is certainly true, but does not distinguish this section from any other sections of the review, which also deal with these themes. To clarify, I found each of these subsections individually very interesting, but I am not sure I agree with their grouping together in a section of their own. In general, I think a broad-scale restructuring would significantly enhance the

manuscript.

We agree that “death regulation” is not an appropriate title for this grouping, which we consider more coherent when framed around models of longevity. We have therefore renamed this section “Models of longevity regulation.”

2. The paper lacks cohesion. Sections do not seem to flow in a very intuitive way, and this can really hamper the reader's progression through the paper. I think this could be remedied quite easily; for example, a simple and impactful solution would be to add a few sentences at the end of each section introduction that mention what the ensuing subsections contain, and suggest how they interrelate.

Thank you for this very appropriate suggestion. This is exactly what we have added in the revised version.

It would also be a nice touch to have subsections within the same section reference each other, so readers can understand how the sections are logically linked. It would also be very simple to edit some of the subsection titles such that they better reflect the section they are in; for example, changing "Stem cell aging" to "DNA Damage as a Driver of Stem Cell Aging", etc.

In the introductory paragraph of each section, while briefly presenting the following studies, we highlighted connections between them whenever relevant.

3. The papers reads as if authors are writing with different scopes for the manuscript in mind, and this makes the manuscript feel inconsistent. My impression based on the abstract and introduction is that the aims of the review are as follows: a) to unify themes from aging research in the context of evolution, development, and ecology, b) to explore diverse aging patterns across species / environments, and the molecular mechanisms underpinning these, and c) to highlight pro-longevity mechanisms across the animal kingdom that may be of relevance for healthy ageing in humans. Many of the subsections do indeed address all of these, but some, for example, read as a "pitch" for a particular model system with little or no reference to the underlying mechanisms driving their particular aging pattern. Some, on the other hand, are quite heavy on mechanistic detail but lack any meaningful reference back to the bigger picture. It would be useful to ensure that all authors are aware of the intended aims such that any sections that diverge too far from these may be revised for consistency.

As we noted above, these discrepancies in length and focus between paragraphs reflect the intended format of a multi-author perspective. To make this clearer we added the name of each author's paragraph.

4. Finally, I appreciate that the authors all work on vastly different systems, on very different scales, and in disparate disciplines. This diversity is of course one of the main strengths of the manuscript. However, as stated previously, the authors need to bear in mind the very broad potential readership of this paper, and ensure that their sections are written such that any biologist could reasonably understand them. For example, given the comparative nature of the paper, some of the more mechanistic subsections should signpost the reader to their taxonomic location; while it is not necessarily of importance that the muscle stem cells discussed are in mice, it is useful for a reader to know that we have moved to mice when it is immediately preceded by a section about planarians. Similarly, some of mechanism-focused subsections would benefit from spending more time introducing some of the molecular players involved so as not to leave behind some of the biologists for whom these names are largely meaningless.

We really appreciate this remark about the strength to combine together a vast spectrum of organisms. To make this clearer we added in the title of the paragraph, when relevant, the name of the organism(s) concerned.

Minor concerns

5. I appreciate the short introductions at the beginning of each section of the review, and find them to be an extremely helpful way to guide the reader through the manuscript with enough background to understand the relevance of each subsection. However, some of the introductions were much shorter than others, and left a lot to be desired; for example, I did not find the "How to regulate death?" introduction very illuminating (although in any case, as I have stated previously, I found the grouping of this section problematic), and also found the "how to better repair DNA" introduction lacking, especially considering how long this section is. It would be useful, for example, to discuss how DNA damage accumulates with age and why, and to explain that most organisms have endogenous repair mechanisms for this, etc. etc. It is also missing a reference for "multiple examples of mutations in DNA repair genes that lead to progeroid phenotypes". In addition, I think that these introductions might be a useful place to outline some other relevant work in the field, as the subsections often focus specifically on the author's own work. This is not a complaint; I understand that breadth must be sacrificed for depth, and there is great merit in describing fine mechanistic details in a review such as this. However, the introductions could be a useful place to reference other relevant work for interested readers.

We agree that more in-depth presentations would be useful for each theme. To address this, we have added references to high-quality reviews in the corresponding introductory sections.

6. I enjoyed the introduction. It was very well-written, providing some original perspectives on the imminent shifts in global demography and ecology and therefore justifying the call for better integration of disciplines of aging research. It gave a very detailed yet accessible background to the molecular and evolutionary biology of aging. However, it would be useful to include some reference to the disposable soma theory of aging alongside the developmental theory, as the "other" leading physiological theory of aging. It feels like something of an omission to exclude it, particularly if this manuscript is picked up by biologists unfamiliar with evolutionary theories of aging.

We are sorry for this omission. We added in the revised version "Alongside these population-genetic frameworks, the disposable soma theory proposes that organisms evolve to allocate limited resources preferentially to reproduction rather than somatic maintenance, leading to progressive decline of repair systems and physiological aging of the body ("soma") (Kirkwood, 1977).".

7. I also very much enjoyed the conclusion, in particular the neat synthesis of the topics of the paper into new questions and insights. However, it did feel like these largely summarised and referred to the sections on the comparative biology of aging, with a focus on non-model systems and the insights to be gained from these, and somewhat glossed over many other more mechanistic sections of the review. It would be nice to see these incorporated into the concluding marks, as they obviously formed a large part of review.

In the revised version, we have incorporated these concluding lessons into the preamble to clarify the overall scope of the review.

8. Considering that many people use reviews to direct research plans, it would be useful to see some more explicit suggestions for future research avenues within each subsection.

At the end of many paragraphs, some suggestions for future research are given.

9. I liked Figure 1, but I am not sure how much it adds to the argument for the use of the Soay sheep as a model for aging in the wild, and I also think that its status as the only figure in the manuscript linked to a particular subsection confers it disproportionate significance. Unless the other authors have figures to add, I am not sure if it is worth retaining.

We agree and have deleted Figure 1

10. I think Figure 2 has a lot going for it. I really like the conceptualisation of these drivers of aging as cogs that can turn to spin the clock of aging faster. I also like the idea that these could be manipulated or pharmacologically targeted to reverse this. However, I am not sure how clear this metaphor is, considering that cogs are often used as symbols of biological mechanisms, and clocks are often used to represent aging - so this at first glance could just look like a cloud of aging mechanisms. I think this would be helped by adjusting the graphics such that the cogs actually interlock, and making the aging clock more integrated into its cog, such that the actual mechanical metaphor is clear. I also think it would be useful to make the brown "reverse aging" arrow more obvious, perhaps with a label (e.g. "pharmacological interventions", or an icon of a syringe, etc.), and perhaps with some other kind of progression arrow coming from the long-lived species, such that it is clear that your implication is that using these insights could lead to anti-aging strategies. I am also not sure if superimposing the diagram over the silhouetted grey age progression in the background is necessary, and I think removing it would reduce clutter in the image. I also think it would help to, at relevant points in the review, mention what these "drivers" of aging are such that the reader is primed for this figure at the end - I think the first time this word is used is in the conclusion.

Many thanks for these insightful suggestions. We have redrawn the figure accordingly.

Additional non-essential suggestions

11. P13 - explain what "thymus involution" is and why it is relevant for aging?

Done

12. P15 - somatic mutations arise frequently in Drosophila intestinal stem cells, and somatic mutation rates across mammals correlate inversely with lifespan - but is there any evidence that these are associated with aging in Drosophila?

Thank you for this suggestion. We now have added the statement that these mutations "increase dramatically during aging".

13. P17 - Capitalise DREAM in the title

Done

14. P19 - DDR - DNA damage response? Define this acronym

Done

15. P20 - IFN-responsive neurons - clarify that this refers to interferon

Done

16. P25 - the final sentence seems like a big jump to go from corals to humans; can you suggest a way in which this research might link to studies on heat waves and telomere length in humans?

There is evidence that heat wave accelerates aging in human. Thus, understanding how telomeres in long-lived corals remain resilient under environmental stress might provide bioinspiration for interventions that mitigate the impact of global warming on

human aging. Although such applications are still very distant, they rest on the same principles as other proposed interventions for healthy aging derived from long-lived species, so we believe it relevant to speculate about them here.

17. P26 - define neoteny (and distinguish it from paedomorphy)

Done

18. P26 - "besides metabolism" - this phrase comes before metabolism has been mentioned for the first time

Done

19. P35 - prolonged presence of gut bacteria - what is the context for this? Is this common, was this an artificial manipulation?

We thank the reviewer for raising this point. In the study cited (Kawamoto et al., 2023), the "prolonged presence of gut bacteria" does not refer to an artificial experimental manipulation. Rather, it reflects the normal, lifelong exposure to commensal microbiota under standard SPF housing conditions. In wild-type mice maintained in SPF environments, aging is associated with the accumulation of senescent germinal center B cells in the ileum. Importantly, this age-associated senescence phenotype is not observed in germ-free mice, indicating that it arises from chronic physiological exposure to commensal bacteria rather than experimental intervention.

Thus, the term "prolonged presence" refers to a common, age-dependent host-microbiota interaction rather than an artificially induced condition.

Dear Eric,

Thank you for sending us the updated version of your article manuscript.

I am pleased to inform you that your manuscript has been accepted for publication in the EMBO Journal as a Perspective.

Your manuscript will be processed for publication by EMBO Press. It will be copy edited and you will receive page proofs prior to publication. Please note that you will be contacted by Springer Nature Author Services to complete licensing and payment information. Please note that as this is invited front-half content, OA charges applicable to this article will be covered.

If you have any questions related, please do not hesitate to contact me.

Thank you again for your kind contribution to The EMBO Journal, which is much appreciated.

with
Best regards,

Daniel

Daniel Klimmeck, PhD
Senior Editor
The EMBO Journal

Please note that it is The EMBO Journal policy for the transcript of the editorial process (containing referee reports and your response letters) to be published as an online supplement to each paper. If you should prefer removal of any referee-only figures included in the point-by-point response(s), e.g. because they may still be used for future publication or because they have been reproduced from published work by others, please do let us know immediately via response email.

More information is available here: <https://link.springer.com/partners/embo-press/editorial-policies#Peer%20review>
